

# A novel virtual reality application for autonomous assessment of cervical range of motion: development and reliability study

Jose Angel Santos-Paz[1], Álvaro Sánchez-Picot[1], Ana Rojo[1], Aitor Martín-Pintado-Zugasti[2], Abraham Otero[1] and Rodrigo Garcia-Carmona[1]

[1] Escuela Politécnica Superior, Universidad San Pablo-CEU, CEU Universities, Madrid, Spain
[2] Departamento de Fisioterapia, Facultad de Medicina, Universidad San Pablo-CEU, CEU Universities, Madrid, Spain

Corresponding author
Jose Angel Santos-Paz,
joseangel.santospaz@usp.ceu.es,
santospaz.ja@gmail.com

## ABSTRACT

**Background**. Neck pain, one of the most common musculoskeletal diseases, affects 222 million people worldwide. The cervical range of motion (CROM) is a tool used to assess the neck's state across three movement axes: flexo-extension, rotation, and lateral flexion. People with neck pain often have a reduced CROM, and they feel pain at the end-range and/or accompany neck movements with compensatory trunk movements. Virtual reality (VR) setups can track the movement of the head and other body parts in order to create the sensation of immersion in the virtual environment. Using this tracking position information, a CROM assessment can be performed using a VR setup that may be carried out autonomously from the user's home. The objectives of this study were to develop a VR experience that could be used to perform a CROM assessment, and to evaluate the intra-rater and inter-rater reliability of the CROM measures guided by this VR experience. To the best of our knowledge, a study of this type has not been carried out before.

**Materials & Methods**. A total of 30 asymptomatic adults were assessed using a VR device (HTC Vive Pro Eye[TM]). Two raters provided support with the VR setup, and the participants were guided by the VR experience as they performed the movements. Each rater tested each subject twice, in random order. In addition to a head-mounted display (HMD), a tracker located on the subject's back was used to measure trunk compensatory movements. The CROM was estimated using only the HMD position and this measurement was corrected using the tracker data. The mean and standard deviation were calculated to characterize the CROM. To evaluate the reliability, the interclass correlation coefficients (ICC) were calculated for intra-rater and inter-rater analysis. The standard error of measurement and minimum detectable change were also calculated. The usability of the VR system was measured using the Spanish version of the System Usability Scale.

**Results**. The mean CROM values in each axis of movement were compatible with those described in the literature. ICC values ranged between 0.86 and 0.96 in the intra-rater analysis and between 0.83 and 0.97 in the inter-rater analysis; these values were between good and excellent. When applying the correction of the trunk movements, both the intra-rater and inter-rater ICC values slightly worsened except in the case of the lateral

flexion movement, where they slightly improved. The usability score of the CROM assessment/VR system was 86 points, which is an excellent usability score.

**Conclusion**. The reliability of the measurements and the usability of the system indicate that a VR setup can be used to assess CROM. The reliability of the VR setup can be affected by slippage of the HMD or tracker. Both slippage errors are additive, *i.e.*, only when the sum of these two errors is less than the compensatory movement do the measurements improve when considering the tracker data.

## INTRODUCTION

Of all the conditions that affect the mobility of the cervical spine, musculoskeletal diseases stand out due to their high prevalence. Musculoskeletal diseases encompass a series of potentially disabling pathologies, including osteoarthritis, rheumatoid arthritis, gout, osteoporosis, traumatic fractures, and sarcopenia (*Hoy et al., 2014*; *Cohen, 2015*). According to the World Health Organization, neck pain is one of the most prevalent musculoskeletal diseases, affecting 222 million people (*World Health Organization, 2021*). Between 30 and 50% of the general population suffer from this condition at some point in their lives, and it is considered the fourth most common disabling pain worldwide (*Hoy et al., 2014*; *Cohen, 2015*). Clinicians rely on the characterization of joints in order to assess the neck's functional state and to design an appropriate treatment plan (*Oliveira-Souza et al., 2020*; *Furness et al., 2018*).

The most commonly-used parameter in clinical practice to assess the function of a joint is its range of motion (*Kisner, 2018*), that is, the number of degrees a joint can move in its various planes of motion. In the neck, this is called the cervical range of motion (CROM) and is measured across three planes that each correspond to a movement axis: the sagittal plane (flexo-extension movement, $X$-axis), the transverse plane (rotation movement, $Y$-axis), and the frontal plane (lateral flexion, $Z$-axis). International guidelines for the management of neck pain describe CROM limitations in people with musculoskeletal diseases that affect the neck, as well as pain at the end-range of active CROM (*Blanpied et al., 2017*).

Another factor to assess when evaluating the condition of a joint is the presence of compensatory patterns (*Van der Kruk et al., 2021*). Compensation is defined as a variation in movement strategy that is used when the normal neuromuscular strategy produces pain or is no longer viable. People with neck pain compensate for their joint conditions by changing the path of movement, using a variety of strategies to achieve functional goals, or by modifying muscle recruitment to perform a movement. Due to redundancies in muscle architecture that result in several muscles being capable of allowing for similar tasks, a specific muscle can compensate for another impaired muscle in order to complete a task

**Table 1  ICC values for the different instruments used to measure CROM in healthy subjects.**

| Study | Instrument | ICC | | | | | |
|---|---|---|---|---|---|---|---|
| | | Intra-rater | | | Inter-rater | | |
| | | Flexo-extension | Right-left rotation | Right-left lateral flexion | Flexo-extension | Right-left rotation | Right-left lateral flexion |
| Oliveira-Souza et al. | CROM device | 0.83 | 0.93 | 0.89 | 0.94 | 0.91 | 0.80 |
| Chalimourdas et al. | Inertial sensors | 0.73 | 0.69 | 0.85 | dnr | dnr | dnr |
| Williams et al. | Goniometer (systematic review) | 0.85 | 0.84 | 0.85 | 0.68 | 0.58 | 0.76 |
| Feng et al. | Optical motion capture | 0.98 | 0.82 | 0.81 | dnr | dnr | dnr |
| | CROM device | 0.94 | 0.79 | 0.79 | dnr | dnr | dnr |
| | Smartphone | 0.91 | 0.73 | 0.91 | 0.89 | 0.77 | 0.85 |
| Ghorbani et al. | CROM device | 0.90 | 0.67 | 0.92 | 0.89 | 0.83 | 0.96 |
| | Inclinometer | 0.76 | 0.68 | 0.80 | 0.82 | 0.83 | 0.77 |
| Law et al. | Electronic goniometer | 0.84 | 0.75 | 0.86 | 0.91 | 0.86 | 0.92 |
| Prushansky et al. | Digital inclinometer | 0.93 | 0.88 | 0.86 | dnr | dnr | dnr |
| Niewiadomski | Opto-electronic acquisition system | 0.81 | 0.83 | 0.86 | dnr | dnr | dnr |
| Raya et al. | Inertial sensors | 0.97 | 0.98 | 0.95 | dnr | dnr | dnr |
| Wolan-Nieroda et al. | CROM device | 0.69 | 0.50 | 0.54 | 0.79 | 0.71 | 0.72 |

**Notes.**

ICC, Intraclass correlation coefficient; CROM device, Cervical range-of-motion device; dnr, did not report.

When the authors only reported the ICC for half movements, an average was made to obtain the ICC of flexion-extension, right-left rotation, and right-left lateral flexion.

without a change in trajectory (*Van der Kruk et al., 2021*). Subjects with neck pain typically compensate by partially performing movements with the trunk instead of just moving the neck.

A wide range of protocols are available to assess the range of motion of the cervical spine. These protocols measure the CROM using different instruments. The CROM device (*Williams et al., 2012*; *Wolan-Nieroda et al., 2020*; *Swinkels & Swinkels-Meewisse, 2014*; *Oliveira-Souza et al., 2020*), inclinometers (*Prushansky, Deryi & Jabarreen, 2010*; *Audette et al., 2010*; *Yoo, Park & Lee, 2011*), and the goniometer (*Williams et al., 2010*; *Luedtke et al., 2020*; *Swinkels & Swinkels-Meewisse, 2014*) are instruments that are typically employed in clinical practice. Among these, the goniometer, due to its small size, portability, and low cost, is the most widely-used solution. However, the intra-rater and inter-rater reproducibility of its measurements, assessed using the intraclass correlation coefficient (ICC), is typically lower than that of the other available instruments (see Table 1).

Different types of motion acquisition systems (*Yoon, Kim & Min, 2019*; *Niewiadomski et al., 2019*; *Cánovas-Ambit et al., 2021*), photogrammetry (*Janjua et al., 2020*; *Baydal-Bertomeu et al., 2007*), and even radiographic images (*Rousseau et al., 2007*; *Ordway et al., 1997*; *Janjua et al., 2020*) have been used in research contexts in order to obtain more accurate and reproducible measurements. However, their higher cost, lack of portability, and sophisticated setups that require a large amount of time to assess a single subject prevent their generalized usage in clinical practice.

New technologies have recently begun to be used for the measurement of CROM. Their aim is to improve the accuracy and reproducibility of the classic instruments'

measurements, but at a low cost and while maintaining portability and simplicity of usage. These include inertial sensors (*Yoon, Kim & Min, 2019*; *Raya et al., 2018*), electromagnetic tracking devices (*Tsang, Szeto & Lee, 2013*; *Tsang, Szeto & Lee, 2014*; *Amiri, Jull & Bullock-Saxton, 2003*), and smartphones (*Chang et al., 2019*; *Ghorbani, Kamyab & Azadinia, 2020*). One of the newest technologies being used for CROM measurement is virtual reality (VR) (*Bechara et al., 2012*; *Kiper et al., 2020*; *Xu et al., 2015*).

A recent systematic review about the validity and reliability of using interactive VR in assessing the musculoskeletal system found limited but promising evidence that VR is a valid and reliable tool to assess range of motion (ROM), but recommended that future studies further investigate VR's psychometric properties. This review showed that the few studies that have used VR to assess CROM used electromagnetic tracking systems as the reference standard to measure ROM, while VR systems were only used to provide a virtual environment that was projected using a head-mounted display (HMD) (*Gumaa, Khaireldin & Rehan Youssef, 2021*). From our point of view, the need for two independent devices when measuring CROM limits the technology transfer possibilities for clinical applicability and hinders the interpretations of the results due to the required synchronization between both devices. It is clear that there is a demand for VR setups that simultaneously provide an immersive experience that guides the user through the movements and measures the CROM without a need for any additional device. Such a VR solution may have the potential for the autonomous assessment of CROM where the VR experience would guide the users throughout the protocol and simultaneously measure CROM, increasing its potential for use in tele-assessment.

The two main objectives of this study were to develop a novel VR application that can perform an autonomous assessment of CROM using only a commercial VR setup, and to evaluate its usability and the intra-rater and inter-rater reliability of its measurements. To the best of our knowledge, this is the first study to look at the intra-rater and inter-rater reliability of a VR setup for CROM assessment. The raters did not give any verbal explanation regarding the movements to be carried out during the assessment, but the VR experience guided the users throughout the protocol. The CROM was calculated using only the measurements of the HMD and the measurements of a tracker placed in the back of the subject to correct the compensatory movements made with the trunk. Both strategies for estimating CROM were compared in order to assess the value of using the additional tracker.

## MATERIALS & METHODS

The protocol followed in this study was approved by the Universidad San Pablo-CEU Research Ethics Committee, (approval code: 549/21/48). After a complete verbal description of the procedures and the purpose of the study, a written informed consent was obtained from the participants.

### Participants

Participants were recruited among students and teachers from the Universidad San Pablo-CEU. The inclusion criteria included men and women of all races between 18 and 65 years

of age without cervical joint pain in the last month and no prior treatment for neck pain. The exclusion criteria were a previous medical diagnosis of: visual impairment that was not corrected with the use of glasses or contact lenses; migraine headache; complex regional syndrome; previous surgeries in the neck and head region; sensory or vestibular alterations; existence of tumors in the craniocervical region; previous fracture in the head or neck region; osseous malformations in the thoracic, cervical, or cranial regions; or idiopathic or otogenic vertigo/dizziness. Following the recommendations used to conduct a reliability study (*Koo & Li, 2016*), 30 subjects were recruited.

## VR setup

VR is defined as "the use of computer-generated virtual environments and the associated hardware to provide the user with the illusion of physical presence within that environment" (*Jayaram, Connacher & Lyons, 1997*). VR setups usually are comprised of an HMD and controllers that allow the user to interact with the virtual environment. HMDs are head-worn devices that project images into the user's field of view while allowing free mobility of the head and, potentially, the entire body (*Rahman et al., 2020*; *Lee, Chang & Park, 2020*). Most HMDs that are currently on the market have displays with high-definition resolution, a large field of view, and high refresh rates. Some VR setups have the option to track additional parts of the body, not just the head (tracked by the HMD) and the hands (tracked by the controllers). These VR setups include trackers that can be attached to the part of the user's body where movement needs to be followed. An example of such a commercial tracker is the HTC Vive Tracker™, which can be attached to objects such as sport equipment, toy weapons, or even body parts (*Niehorster, Li & Lappe, 2017*).

Since the introduction of VR in the world of computing, it has been used for different purposes, including gaming, education, industrial applications, and medical applications. In the field of medicine, VR has been used for training medical procedures (*Tin, Hertelendy & Ciottone, 2021*; *Moon et al., 2021*; *Nambi et al., 2021*), to evaluate sight (*Yasuda et al., 2020*; *Sabu et al., 2020*), and in rehabilitation (*Chen, 2021*; *Laver et al., 2015*; *Tin, Hertelendy & Ciottone, 2021*).

VR provides a virtual environment that immerses its users. Presence is the sensation that the user feels when truly immersed in a virtual world (*Rubin, 2018*), *i.e.*, the subjective part of the experience. Immersion is the objective part of the experience, where the visualization of the images, frames projected per second, audio stimuli, and kinesthetic stimuli (*Găină et al., 2021*) are taken into consideration. VR tracks the users' neck movements with high accuracy to provide better immersion. Therefore, we hypothesize that we can take advantage of this precise tracking to measure CROM. Additionally, through the usage of tracker devices, the movement of any other body part, such as the trunk, can also be tracked. This could permit the identification of compensatory movements.

Our measuring instrument was a VR setup that was comprised of one HTC Vive Pro Eye™ HMD, one Vive Tracker™, two wireless HTC Vive Controllers™, and two Lighthouse™ v2 base stations. The HMD has a resolution of 1,440 × 1,600 pixels per eye, a refresh rate of 90 Hz, a field of view of 110°, and built-in Hi-Res Certified headphones.

The HMD uses room-scale tracking technology to interact with the virtual environment using Valve's SteamVR$^{TM}$ Lighthouse system. The Lighthouse base stations contain infrared LEDs and a laser array that sweeps in the horizontal and vertical directions. The surface of the HTC Vive$^{TM}$ devices (the HMD, controllers, and trackers) has photodiodes that measure when the laser has hit them. Knowing the time differences at which the different photodiodes are hit by the laser enables the accurate tracking of the three-dimensional position and orientation of the devices. One of the advantages of this tracking system is that it provides six degrees of freedom that track both the translation and rotation of the user within the virtual environment. To calculate the CROM using an HMD, we used the rotation axes. The movement in the sagittal plane corresponds to flexion-extension ($X$-axis), the movement in the transverse plane corresponds to right-left rotation ($Y$-axis), and the movement in the frontal plane corresponds to right-left lateral flexion ($Z$-axis) (see Fig. 1).

A tracker was used to follow the movement of the subjects' backs in order to measure trunk compensations. It was attached to the back of the subjects over the fourth thoracic spine (T4), since this sensor placement has shown good reliability in previous research (*Raya et al., 2018*; *Tsang, Szeto & Lee, 2014*) and may prevent collisions with the HMD during cervical extension movement. When placing the tracker on the subject's back, they were asked to stand in a T-pose position. The T4 spinous process was identified by palpating the T3 spinous process (aligned with the spine of the scapula) (*Han et al., 2012*) and moving one segment immediately inferior. The tracker has a blue triangle drawn on its center, and one of the sides of the triangle was placed over T4. This tracker position also ensures that the tracker has a large contact surface with the back, minimizing any movement or slippage (see Fig. 2A). To attach the tracker, we used a custom-built X-shaped harness formed by an elastic band and a 3D printed adapter (see Fig. 2). The elastic band is 2 m long and 20 mm wide, with a zigzag one mm flexible silicon stripes to prevent slippage (see Figs. 2B–2C). The silicone stripes were on the side of the strap that was in contact with the subject to prevent the displacement of the tracker. The harness consists of two handles placed in such a way that each of them was on one arm, at shoulder height.

The handheld controllers were used to track the position of the hands and control the application using the trackpad and trigger buttons. The tracking of the hand's position was used to increase the sensation of immersion, since in the virtual environment, the avatar representing the user has hands that replicate the hands' position in the real world. Although all devices were tracked, only the data from the HMD and the tracker (to identify potential compensatory movements) were stored for CROM assessment. The stored data contained the three-dimensional position and rotation of both devices sampled at a fixed rate of 90 Hz, which is the maximum fixed-interval rate provided by the VR setup. The game engine updated and recorded the timestamped position and orientation in a CSV file to be processed later. Since OpenVR was used, this sampling rate had to match the framerate of the VR device used (HTC Vive Pro Eye$^{TM}$).

To guarantee that the measurements were completely blind between raters, it was decided that each rater use an independent computer. The computer used by rater A was a HP OMEN Intel Core i7-10750H with 16GB of RAM and an 8GB NVIDIA GeForce RTX

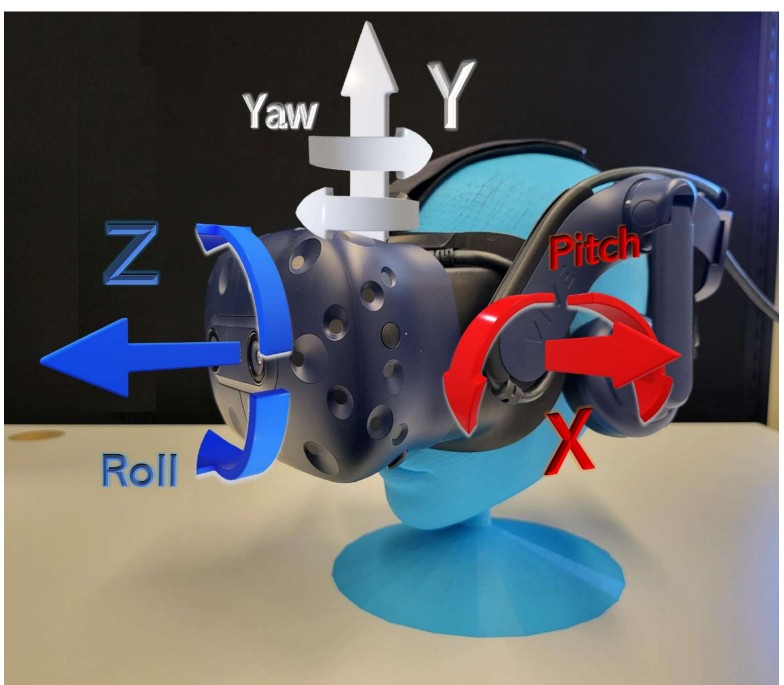

**Figure 1 HMD degrees of freedom.** The HMD in the virtual environment had six degrees of freedom, corresponding to movements in the $X$-axis, $Y$-axis, and $Z$-axis, and rotation in each axis (pitch, yaw, roll, respectively).

2070 Super GPU, running the 64-bit version of Windows 10 Pro, SteamVR[TM] 1.16.10, and the CROM assessment application described in the next section. The computer used by rater B was a DELL G5 5590 Intel Core i7- 9750H with 16GB of RAM and a 6GB NVIDIA GeForce GTX 1660 Ti GPU, running the 64-bit version of Windows 10 Home, SteamVR[TM] 1.17.16, and the CROM assessment application. Both computers were powerful enough to run the VR experience at least at 90 frames per second, with a refresh rate that was supported by the HMD without missing frames.

## VR application

We developed an application to guide the subjects while they performed the movements necessary for CROM measurement. The VR application was developed using the Unity 3D game engine (v.2019.4.19f1) with OpenVR and the Steam framework for HTC Vive.

The VR application for CROM assessment was made up of two interfaces or viewpoints: the VR experience that is seen by the user through the HMD, and an interface shown in the monitor of the computer that only the rater observes (see Fig. 3A). The rater interface has a menu in which a number uniquely associated with the subject that is being evaluated is entered; this number is used to identify the evaluation sessions that correspond to that particular subject. The rater interface also has control buttons that allow for starting and stopping the assessment and exiting the application.
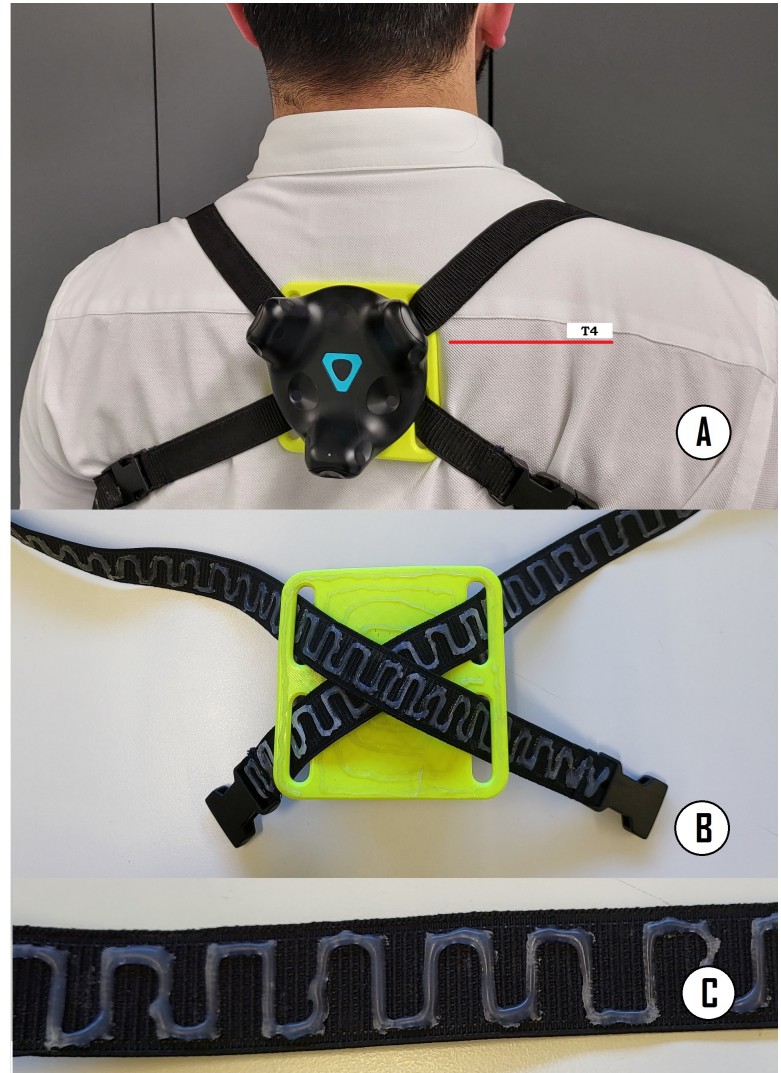

**Figure 2** (A) The tracker used to correct compensations was attached to the back of the subjects over T4, (B) X-shaped harness formed by an elastic band and a 3D printed adapter, and (C) flexible silicone zigzag stripes were added on the side of the strap that was in contact with the subject to prevent slippage of the tracker.

The virtual environment is a medical/physiotherapeutic office (see Figs. 3A–3E). It is sparse in order to avoid distractions during the movements. One of the more important considerations when designing the room was to avoid clear lines that crossed the whole ceiling since they could serve as a guide for the subjects during movements and influence the trajectory of the movements or their range.

The virtual room contained a desk, two chairs with a human figure sitting in one of them, another chair where the avatar of the subject is sitting, and some other medical furniture placed further away. The figure is placed in front of the subject (see Figs. 3B–3E). This figure performs, as a way of example, the movements that the subject will need to

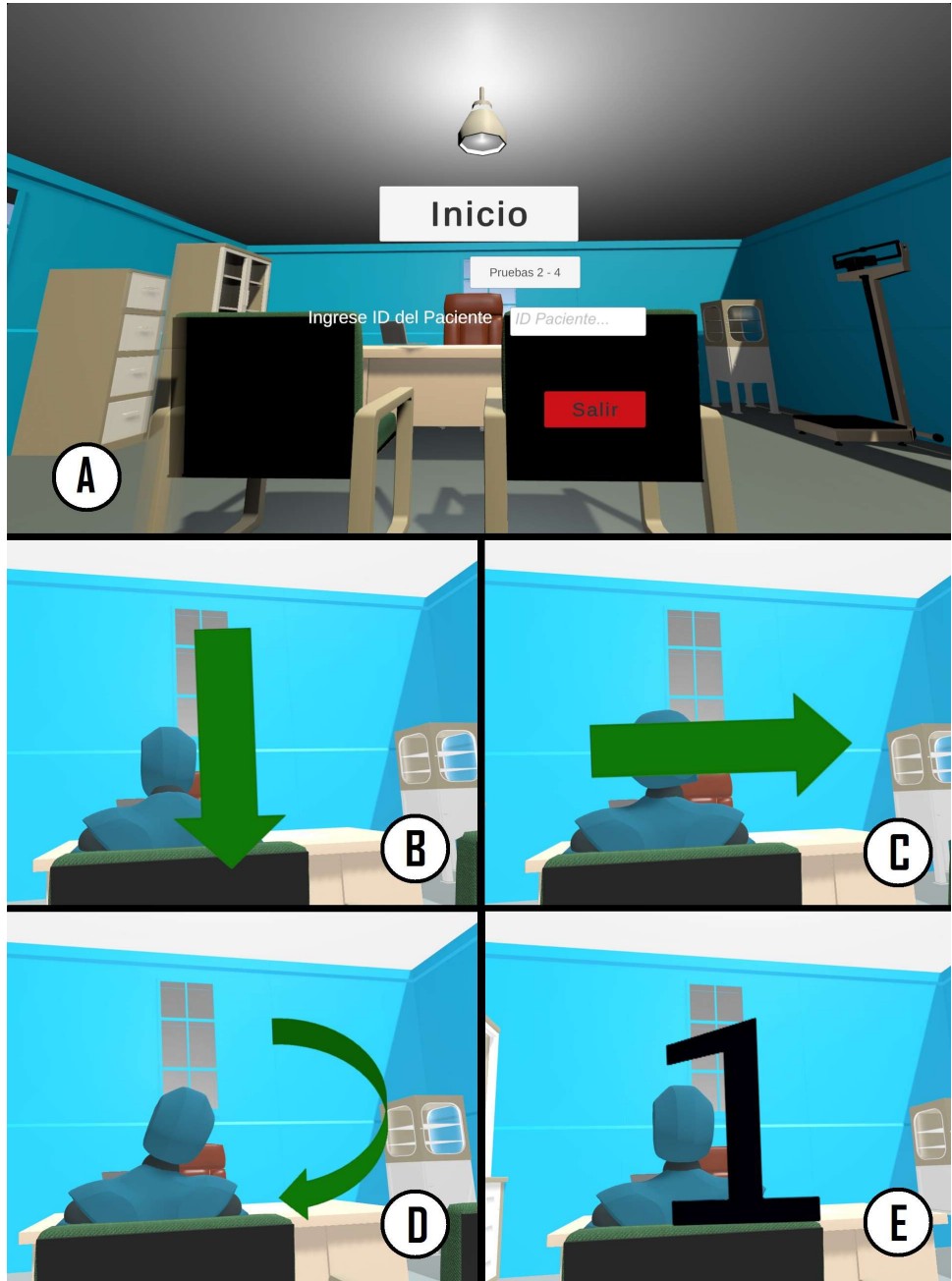

**Figure 3 Graphic interface of the CROM assessment application.** (A) Application start menu, only observed by the rater. (B, C and D) Example of the instructions for the flexion-extension, rotation, and lateral inclination movements observed by the subject when performing the measurement. (E) Example of the numbers that indicate the repetition number being executed.

carry out. Audio instructions played by headphones built in the HMD also guide the user through the process and specify the movements that need to be performed. At the beginning of these instructions, arrows visually indicate the direction of the movements (see Figs. 3B–3D). When performing the movements, a large black number in the center of

the field of view indicates what repetition the subject is in (see Fig. 3E). Using the inverse kinematic solver in Unity, the head and hands of the virtual avatar are synchronized with the person wearing the HMD and the controllers. Thus, the user can view the virtual body from the first-person perspective, enhancing the sense of embodiment or body ownership.

As shown in the application flowchart (see Fig. 4), during the first measurement, the VR experience reproduces an audio clip with a brief description of the objective of the evaluation, as well as detailed indications of the movements that the subject has to perform. Then the experience guides the subject through warm-up exercises similar to the movements that are performed for CROM assessment, which are also exemplified by the human figure. In addition to the warm-up series, the subject must perform three repetitions of each movement. Each repetition is indicated by a large number that appears in the center of the screen (see Fig. 3E). At the end of the third repetition of a given movement, the explanation of the next movement begins. Once the three movements have been completed, a message informing the user that the evaluation session has ended is played. After the first measurement, the VR experience only provides short instructions about the movements, since the complete detailed instructions were already presented during the first measurement. Since the subject has already performed one warm-up series and, at least, a complete series of movements for CROM assessment, the warm-up series is not performed during the subsequent measurements. Similar to the first measurement, each repetition of each movement is indicated with a large number that appears in the center of the screen and is illustrated by the figure. At the end of the third repetition of each movement, the explanation of the next movement begins.

The tracking data from the HMD and tracker have a sampling rate of 90 Hz. They were stored in a CSV file as follows: sample number, timestamp, device ID, X, Y, Z, qW, qX, qY, qZ, aX, aY, and aZ. The timestamp follows the HH: mm:ss.ffff format. The device is identified with a single letter: H for the HMD and T for the tracker. X, Y, and Z correspond to the device position in the space; qW, qX, qY, and qZ to the quaternion; and aX (Pitch), aY (Yaw), and aZ (Roll) are the Euler angles (see Fig. 1) (*Kong, 2014*). To enable the reproducibility of the results of this study, as well as to enable other studies to analyze the data collected, a transcript of the audio message played in the VR application in both its long and short versions, and the 120 CSV files corresponding to each of the four sessions of the 30 subjects are provided as Supplemental Material (see File S1).

## Experimental protocol

Before starting the measurements, the physical area was prepared. A 2 × 2 meter room area was physically delineated and framed on the floor using masking tape before starting the assessment of any subject. In this way, the virtual room used in all of the sessions corresponded to the same physical space. As shown in Fig. 5, the base stations were placed at opposite corners of this area on top of tripods with their positions also marked on the floor. The chair in which the subject was seated was placed at the center of this area and its position was also marked on the floor to ensure that the same location was used for all the subjects. The computers of rater A and rater B were in front of the subject chair to facilitate changing the computers between measurements.

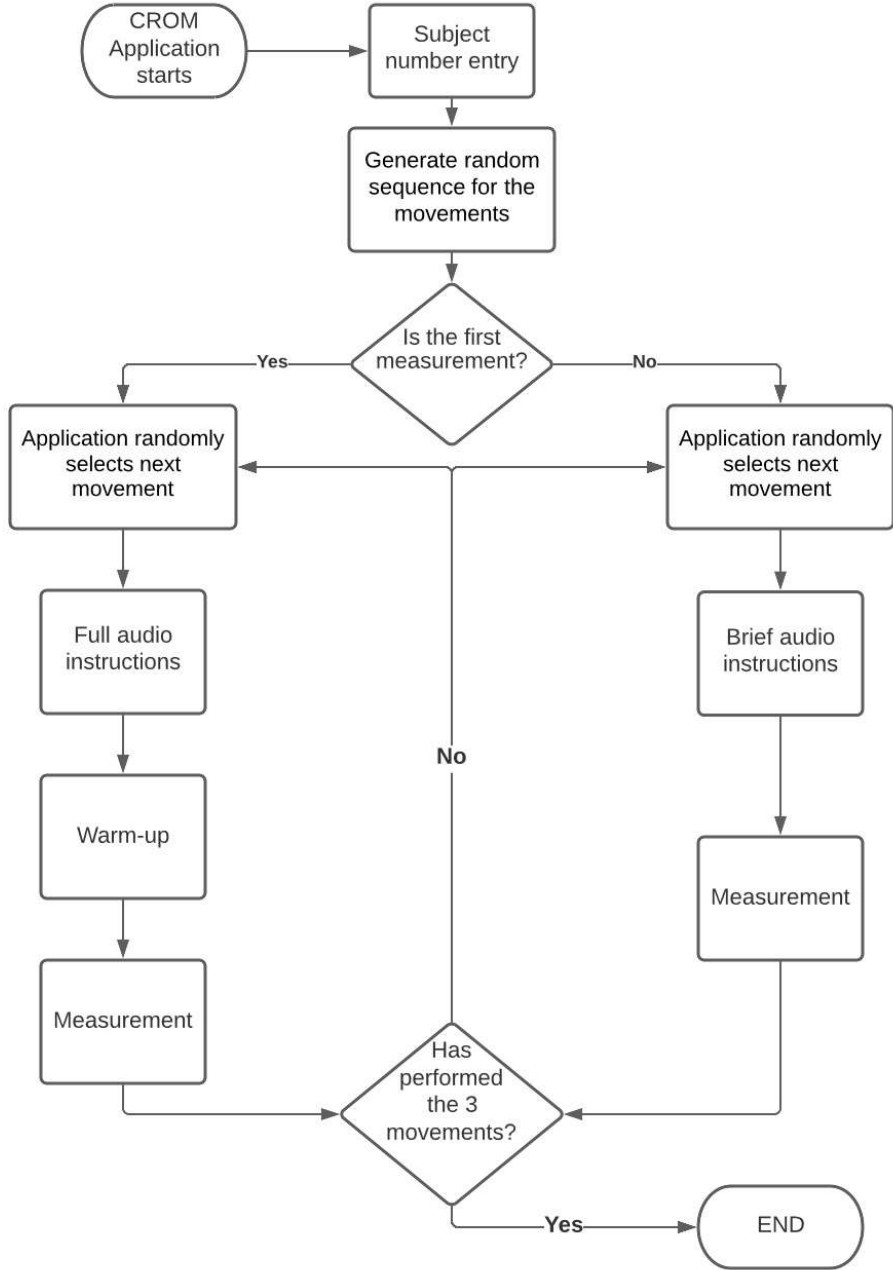

**Figure 4** **Flowchart of the CROM assessment application.**

In SteamVR^TM, the room setup is the procedure in which the limits of the physical area where the interactions in the virtual environment can take place are established. This configuration only needs to be done once, as long as the position and orientation of the base stations are not changed. If the base stations are moved or their orientation is changed, the room setup must be carried out again. The space used for the measurements was also used for other purposes and it was necessary to remove the base stations at the conclusion

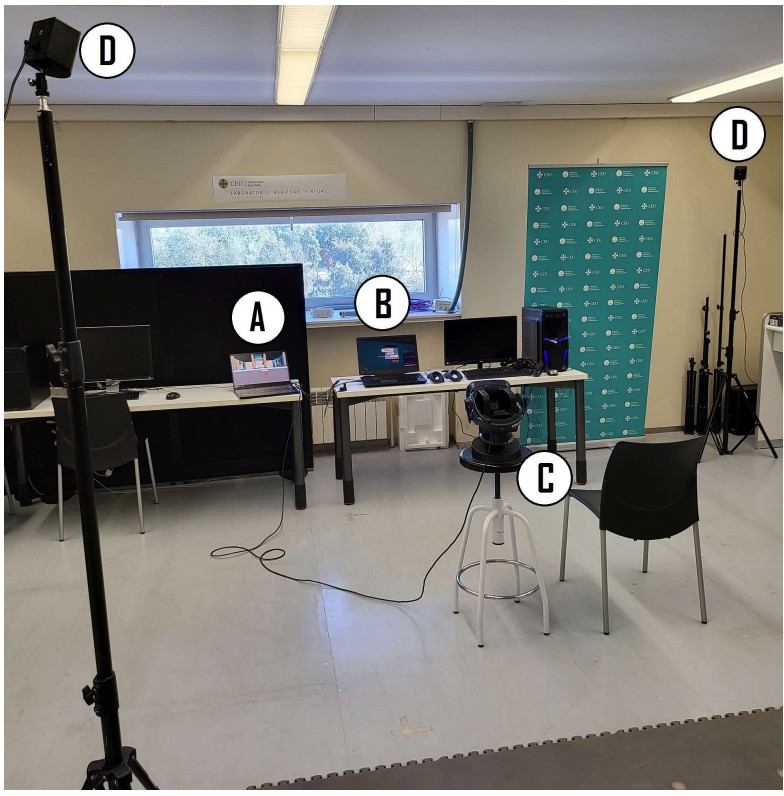

**Figure 5 Measurement area.** (A) Computer of rater A. (B) Computer of rater B. (C) Subject location and HMD. (D) Base stations.

of each measurement session. Hence, to ensure that the measurements were taken within the same area every time, a room setup procedure was performed at the beginning of each measurement session for each of the two computers. Between three and five individuals were assessed during each session. During the room setup, the tape marks of the delineated floor area were used as guidelines to generate a 2 × 2 meter virtual room. If the mapped area was not equal to 2 × 2 m, it was manually edited in SteamVR to obtain the desired size. In all cases, the virtual room was oriented in such a way that the virtual chair faced the computers, since this was also the position of the real chair in which the subject was seated.

Two evaluators (rater A and rater B) supported the subjects when using the VR equipment and controlled the CROM assessment application from their respective computers. Both raters are engineers with experience using and developing VR applications and are familiar with the CROM assessment application. The rater who carried out the first assessment for each subject was selected randomly based on a sequence of random numbers generated with a computer. While an assessment was being performed by one rater, the other waited outside the room.

The measurement protocol of this study was made up of four CROM assessments for each subject, alternating between raters. Before starting the first assessment, the randomly

chosen rater explained to the subject the operation of the VR application and the correct use of the controllers. No instructions about how to perform the movements for CROM assessment were given by the raters at any point; all these instructions were provided by the VR experience. The raters limited their explanations and their support to the usage of the VR equipment and the placement of the tracker on the back of the subject. In accordance with the current COVID-19 health protocols, during the experiments the subjects wore face masks and were also provided with disposable eye masks to prevent their facial skin from coming into direct contact with the HMD.

The subjects could use either hand to control the application with the VR controllers. By pressing any trackpad, the subject was placed in the VR experience behind the figure that illustrated each movement to be performed. The trigger button was used to start the measurements. The order of the movements that each subject must carry out followed a random sequence generated automatically by the computer. Each sequence contained the movements of flexion-extension, rotation, and lateral flexion in a random order without any of them being repeated within the same sequence.

The first rater started the VR application for CROM assessment. The display on the computer screen indicated that the first evaluation is going to be carried out, and a warm-up and detailed instructions on how to perform the movements was given. During the assessment, the raters oversaw the subject without intervening. At the end of the measurement, the rater removed the HMD and tracker from the subject. The first rater then left the room and the other rater entered the room to start the second evaluation. During the second assessment, there was no warm-up stage and the VR experience provided only abbreviated instructions on how to perform the movements. The third and fourth assessments were carried out in a similar way to the second, including the removal of the HMD and tracker between them.

## CROM calculation

To perform the CROM calculation, we used Matlab R2020a. Each subject's data were processed individually. The CROM assessment VR application stored the HMD and tracker's pitch, yaw, and roll, which corresponded to flexion-extension, rotation, and lateral flexion, respectively. The angles recorded from the VR setup were in the range (0–360). Using a wrapping function, we mapped them to the interval ($-180$–$180$). In this new interval, the 0-degree position corresponded to the head of the subject when in a neutral position before starting the movement where the helmet was perfectly aligned with the subject's head. Under this assumption, in the flexion-extension movement, the positive angles corresponded to the flexion and the negative to the extension. For rotations, the positive angles corresponded to the right rotation and the negative to the left rotation. For the lateral flexions, the positive angles corresponded to the left lateral flexion and the negative to the right lateral flexion. In practice, there will be some misalignment between a vector normal to the subject's head and a vector normal to the helmet, which will result in a small (positive or negative) angle in the neutral position, which needs to be corrected.

To calculate the CROM for each movement, we started by plotting the corresponding angle recorded by the HMD versus time. No filtering or any other type of signal conditioning

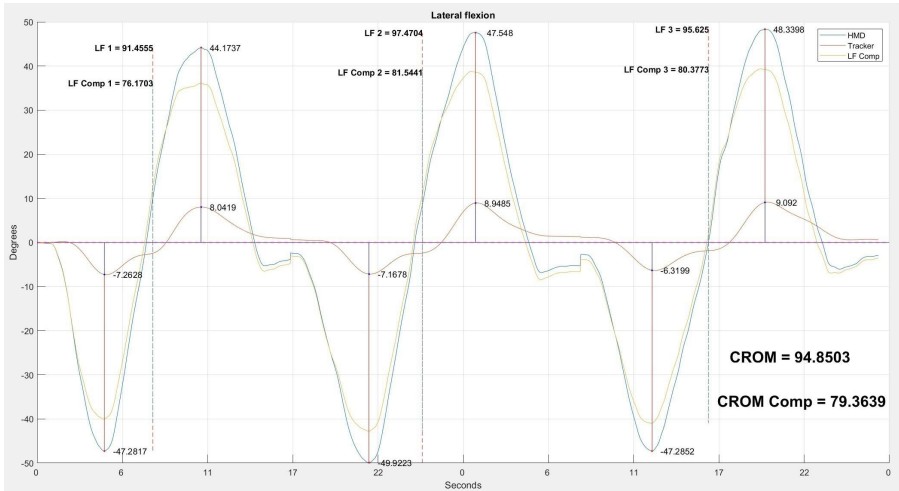

**Figure 6** **Three repetitions of the lateral flexion movement.** Example of the analysis carried out with the three repetitions of the lateral flexion movement. The blue line shows the head movement captured by the HMD. The red line shows the compensatory movement of the back captured by the tracker. The orange line shows the movement resulting from removing the compensation of the back from the movement of the head. For each repetition, we show the maximum and minimum values. The CROM and corrected CROM, calculated as the average value of the three movements, are shown.

was applied to the time series since it was not deemed necessary. Visually, the positions where the three repetitions of the movement were identified, and we manually selected a time position slightly before the first repetition in which the subject's head was in a neutral position. This position was approximate and small variations in its identification should not affect the final result of the measurement. From this selection, the average value of the first 10 measurements was calculated; this value corresponded to the estimate of the misalignment between the helmet and the subject's head in a neutral position. This average was subtracted from each angle measurement to correct the misalignment. Next, we manually selected a time position slightly before the start of each repetition, and another time position slightly after the end of each repetition (see Fig. 6).

The maximum and minimum peaks of each repetition were identified using the manually selected start and end times. The CROM value of each repetition was calculated as the difference between the maximum and the minimum of the movement. Following the same approach used in *Romero Morales et al. (2017)* and *Raya et al. (2018)*, we calculated the average CROM value of the three repetitions, obtaining the final value for the flexion-extension, rotation, and lateral flexion when the angle being processed corresponded to pitch, yaw, and roll, respectively. These were the measurements without the correction of the compensatory movements made with the trunk.

The temporal positions that were manually selected for the calculation of the angle corresponding to the neutral position and the beginning and end of each one of the three movements, were stored. These same positions were used for the calculation of the CROM with the correction of the compensatory movements. To this end, the value of the angle recorded by the tracker at each instant was subtracted from the angle recorded by the

HMD. The resulting angle corresponded to the movement made only with the head. The compensation-corrected CROM measurement was then calculated over the corrected angle time series using the temporal positions stored during the calculation of the CROM without this correction. The person in charge of the data processing described in this section was one of the engineers who also acted as a rater.

## Effect of compensation correction on the measurement error

In this study, we hypothesized that the use of the tracker to correct trunk compensations should increase the reliability of the measurements. However, it is possible that the reliability added by the correction does not compensate for the measurement error accumulated using multiple instruments. Let us represent the measurements of the HMD and the tracker as:

$$ROM_{HMD} = CROM + Trunk\_compensation + Measurement\_error \tag{1}$$

$$ROM_T = Trunk_{compensation} + Measurement\_error \tag{2}$$

where $ROM_{HMD}$ is the measurement of the HMD; $ROM_T$ isthe measurement of the tracker; $CROM$ is the movement performed with the neck; $Trunk\_compensation$ is the compensation movement made by the subject's trunk; and $Measurement\_error$ corresponds to the HMD and tracker's measurement errors which, for the sake of simplicity, are considered to be of equal magnitude for both devices. Within the measurement error, we included the error inherent to each measuring instrument, which in this case is given by the tracking accuracy of the VR setup (*Passos & Jung, 2020*), plus any possible user/procedure error. We believe that the instrument error is dominated by this second source and is mainly caused by the slippage of the HMD or the tracker relative to the head and back of the subjects, respectively. When the compensation correction is applied to the HMD measurements, the instrument errors are not canceled out, but instead increased due to error propagation:

$$CROM_C = ROM_{HMD} - ROM_T = CROM + 2Measurement\_error \tag{3}$$

where $CROM_C$ is the CROM measurement with compensation correction. Therefore, if *Trunk_compensation* is less than *Measurement_error*, the error of the $CROM_C$ measurement is greater than the error of the measurement obtained directly from the HMD without applying any compensation correction (see Eqs. (1) and (3)). A greater reliability of the corrected measures with respect to the uncorrected ones would indicate that Measurement_error is less than Trunk_compensation, while a lower reliability would point towards the opposite.

## Statistical analysis

The statistical analysis was performed using version 27 of the IBM SPSS Statistics statistical package. Since our sample size was less than 50, the Shapiro–Wilk test was used as a test of normality (*Ahmad & Khan Sherwani, 2015*). The level of statistical significance was established as $p < 0.05$. Subsequently, the mean and standard deviation (SD) were calculated to characterize the CROM obtained in each of the three movement axes for the different subjects.

The ICCs were calculated, and the ICC (3, k) model was used for intra-rater analysis, and the ICC (2, k) model was used for inter-rater analysis (*Koo & Li, 2016*). Reliability aims to measure the coherence between repeated measurements of the same variable in the same individual that is subject to the same conditions. While intra-rater reliability deals with the consistency of a single rater evaluating the same event on two different occasions, inter-rater reliability focuses on measuring the variability of two or more raters evaluating the same event on the same occasion (*Koo & Li, 2016*). ICC values from 0 to 0.5 are considered poor, between 0.5 and 0.75 considered moderate, between 0.75 and 0.9 considered good, and above 0.90 considered excellent (*Koo & Li, 2016*).

The standard error of measurement (SEM) and minimum detectable change (MDC) were calculated (*Portney, 2020*). The SEM can be estimated using the ICC:

$$SEM = S_P \sqrt{1 - ICC}$$

where $S_P$ is the pooled standard deviation of measurements. MDC is useful to characterize the effectiveness of a device to detect changes in the measured variable and can be calculated using the SEM. When computed at the 90% confidence level (*Kolber et al., 2011*; *Marszałek & Molik, 2019*; *Laird, Kent & Keating, 2016*), MDC is given by:

$$MDC_{90} = 1.65 x SEM x \sqrt{2}.$$

## System usability

To measure the usability of the VR system as a CROM assessment tool, we used the Spanish version of the System Usability Scale (SUS) (*Sevilla-Gonzalez et al., 2020*). The SUS is a quick and easy-to-use tool for both system users and researchers. It provides a single score on a scale that is easily understandable (*Bangor, Kortum & Miller, 2008*). The SUS questionnaire consists of 10 questions designed to be answered by the user after interacting with the system. The questions alternate between positive and negative statements to avoid respondent bias. Each statement can be scored with a value between 1 and 5, where 5 is "Strongly agree" and 1 is "Strongly disagree". The odd questions are positive statements, and they are scored by subtracting one point from the rating of the user. The even questions are negative statements, and they are scored by subtracting 5 from the rating of the user. When adding all of the scores, a scale of 0–40 is obtained which, when multiplied by 2.5, becomes a scale of 0–100 (*Brooke, 1995*). For a system to have good usability, it must obtain a result between 68 and 84 points. If the system obtains a score higher than 85 points, it is considered to have excellent usability (*Sevilla-Gonzalez et al., 2020*).

When all four CROM assessments were finished, the subjects were asked to anonymously fill out the SUS questionnaire to evaluate the VR experience.

## RESULTS

A total of 30 asymptomatic, mostly young adult Caucasian subjects from a university community participated in the study: 19 men with an average age of 33 ± 11 years, and 11 women with an average age of 31 ± 10 years. Table 2 presents the anthropometric information of the subjects.
**Table 2 Anthropometric data of the subjects.**

| Subjects | Number | Age [years] | Height [m] | Weight [kg] | BMI [kg/m$^2$] |
|----------|--------|-------------|------------|-------------|----------------|
| Male | 19 (63%) | $33.05 \pm 10.73$ | $1.76 \pm 0.09$ | $75.82 \pm 12.94$ | $24.20 \pm 2.43$ |
| Female | 11 (37%) | $30.73 \pm 10.45$ | $1.64 \pm 0.05$ | $62.73 \pm 10.87$ | $23.55 \pm 5.03$ |
| TOTAL | 30 (100%) | $32.20 \pm 10.50$ | $1.72 \pm 0.10$ | $71.02 \pm 13.60$ | $23.96 \pm 3.54$ |

Notes.
BMI, Body mass index.
Values are presented as mean $\pm$ standard deviation.

**Table 3 CROM measured with and without compensation.**

| | Without compensation correction | With compensation correction |
|---|---|---|
| Rater A | | |
| Flexion-extension [°] | $111.93 \pm 20.49$ | $103.48 \pm 15.62$ |
| Right-left rotation [°] | $139.62 \pm 18.90$ | $134.45 \pm 15.81$ |
| Right-left lateral flexion [°] | $88.77 \pm 14.82$ | $78.16 \pm 10.56$ |
| Rater B | | |
| Flexion-extension [°] | $110.99 \pm 19.40$ | $102.81 \pm 15.25$ |
| Right-left rotation [°] | $141.17 \pm 18.71$ | $135.22 \pm 15.17$ |
| Right-left lateral flexion [°] | $88.76 \pm 13.78$ | $77.98 \pm 10.24$ |

Notes.
CROM measured without compensation and with compensation, organized by rater. Values presented as mean $\pm$ standard deviation.

The Shapiro–Wilk test indicated that the distributions of the CROM movements were normal in all cases except one ($p$-value $> 0.05$). In the case of lateral flexion without compensation, the presumption of having a normal distribution (value of $p < 0.05$) was not fulfilled. This was due to the existence of an outlier corresponding to a single test subject whose extremely high lateral flexion measurement was probably affected by some artifact. After the removal of this subject's lateral flexion measurement, the Shapiro–Wilk test indicated that normality was met for all CROM movements.

Table 3 shows the mean and SD of the measurements obtained with and without the compensation correction for each of the raters. The compensation was greater in lateral flexion: there was a difference of 10.61° for rater A and 10.78° for rater B.

The intra-rater and inter-rater reliability results are shown in Tables 4 and 5, respectively. The intra-rater analysis for the measurement without compensation taken into account yielded ICC values that ranged from 0.86 to 0.96, while the values for the measurement with the compensation corrected ranged from 0.79 to 0.96. In both cases, the lowest values corresponded to the flexion movement measured by rater A. Regarding the inter-rater reliability of the CROM, the ICC values for the measurement without compensation ranged between 0.81 and 0.97, while the values for the measurement with compensation varied between 0.82 and 0.97.

The SEM and MDC$_{90}$ in the measurements with compensation correction were in all cases lower when compared to the measurements without, although the difference was mostly small (Tables 4 and 5). Regarding the usability of the VR system as a CROM

**Table 4   Intra-rater reliability analysis.**

| Intra rater | Without compensation correction | | | | With compensation correction | | | |
|---|---|---|---|---|---|---|---|---|
| | ICC | CI 95% | SEM [°] | MDC$_{90}$ [°] | ICC | CI 95% | SEM [°] | MDC$_{90}$ [°] |
| **Rater A** | | | | | | | | |
| Flexion-extension | 0.86 | [0.71 0.93] | 7.69 | 17.93 | 0.79 | [0.56 0.90] | 7.22 | 16.84 |
| Right-left rotation | 0.93 | [0.85 0.97] | 5.18 | 12.10 | 0.93 | [0.84 0.96] | 4.36 | 10.18 |
| Right-left lateral flexion | 0.89 | [0.77 0.95] | 4.91 | 11.46 | 0.92 | [0.83 0.96] | 3.03 | 7.07 |
| **Rater B** | | | | | | | | |
| Flexion-extension | 0.94 | [0.86 0.97] | 4.98 | 11.63 | 0.92 | [0.82 0.96] | 4.48 | 10.45 |
| Right-left rotation | 0.96 | [0.91 0.98] | 3.77 | 8.79 | 0.96 | [0.91 0.98] | 3.17 | 7.39 |
| Right-left lateral flexion | 0.91 | [0.81 0.96] | 4.17 | 9.73 | 0.94 | [0.87 0.97] | 2.55 | 5.95 |

**Notes.**
ICC, Intraclass correlation coefficient; CI, Confidence interval; SEM, Standard error of measurement; MDC, Minimum detectable change.

**Table 5   Inter-rater reliability analysis.**

| Inter rater | Without compensation correction | | | | With compensation correction | | | |
|---|---|---|---|---|---|---|---|---|
| | ICC | CI 95% | SEM [°] | MDC$_{90}$ [°] | ICC | CI 95% | SEM [°] | MDC$_{90}$ [°] |
| **First Measurement** | | | | | | | | |
| Flexion-extension | 0.83 | [0.64 0.92] | 8.17 | 19.07 | 0.82 | [0.63 0.92] | 6.17 | 14.41 |
| Right-left rotation | 0.93 | [0.85 0.97] | 5.09 | 11.88 | 0.92 | [0.83 0.96] | 4.47 | 10.44 |
| Right-left lateral flexion | 0.81 | [0.60 0.91] | 6.32 | 14.75 | 0.88 | [0.75 0.94] | 3.63 | 8.47 |
| **Second Measurement** | | | | | | | | |
| Flexion-extension | 0.97 | [0.93 0.98] | 3.72 | 8.67 | 0.96 | [0.92 0.98] | 3.14 | 7.32 |
| Right-left rotation | 0.97 | [0.94 0.97] | 3.16 | 7.38 | 0.97 | [0.93 0.98] | 2.86 | 6.67 |
| Right-left lateral flexion | 0.95 | [0.90 0.98] | 3.15 | 7.36 | 0.96 | [0.91 0.98] | 2.19 | 5.10 |

**Notes.**
ICC, Intraclass correlation coefficient; CI, Confidence interval; SEM, Standard error of measurement; MDC, Minimum detectable change.

assessment instrument, we obtained an overall score of 86 with a standard deviation of 11. There were no reports from the subjects of any adverse events associated with the VR experience.

## DISCUSSION

There is no absolute consensus on normal CROM values in the scientific literature, since CROM is affected by factors such as the age or gender of the participants, which hinder the reproducibility of studies. However, several authors have reported normal CROM values (*Raya et al., 2018*; *Sarig-Bahat, Weiss & Laufer, 2009*; *Oliveira-Souza et al., 2020*; *Williams et al., 2012*; *Sukari et al., 2021*; *Niewiadomski et al., 2019*; *Swinkels & Swinkels-Meewisse, 2014*; *Amiri, Jull & Bullock-Saxton, 2003*; *Furness et al., 2018*; *Sarig-Bahat, Weiss & Laufer, 2009*). Our results, considering the standard deviation, are consistent with these previous reports. The most similar study to ours with which we can compare our results is the one performed by *Kiper et al. (2020)*, where VR was used as part of a CROM measurement protocol, although their measurements were taken by an external 3D motion-tracking system. This study evaluated 35 healthy subjects and presented results for flexion, extension,

left rotation, right rotation, left lateral flexion, and right lateral flexion movements. Following the CROM assessment methodology proposed in *Raya et al. (2018)*, our study evaluated the complete movements of flexion-extension, right-left rotation, and right-left lateral flexion. We calculated the values of full movements from the half movements used in this study, and obtained values of 125.90° ± 14.99 in flexion-extension, 148.32° ± 15.78° for right-left rotation, and 77.99° ± 10.56 for right-left lateral flexion, respectively, when using an immersive VR setup, and 117.72° ± 14.18, 145.79° ± 11.25, and 79.14° ± 8.29, respectively, when a non-immersive VR setting was used. Considering the confidence intervals, these results are compatible with those presented in Table 3.

Other authors have used VR within a CROM measurement protocol (*Bechara et al., 2012*; *Kiper et al., 2020*). However, they did not use it as a measurement instrument. Instead, they employed some other sensor for the measurement, and the VR equipment was used just to guide the subjects through the required movements. From a methodological point of view, the work most similar to ours is *Xu et al. (2015)*, since the authors did use the position information of a HMD to measure the motor range of the neck. This study involved 10 healthy subjects and compared CROM measurements obtained with the HMD and an inertial sensor mounted on the HMD. However, this work focused on evaluating the consistency of the HMD measurements with those of the inertial sensor, and not in a proper CROM assessment. The participants did not perform the flexion-extension, rotation, and lateral flexion movements up to the maximum active range; they just had to perform them until they reached a series of targets that were displayed on the HMD. Therefore, the values reported in that work (69.8°±7.7, 99.5°± 5.9, and 16.2°± 3.6° for flexion-extension, rotation, and lateral flexion, respectively) were much lower than the maximum range values of the healthy subjects.

Tables 4 and 5 show that the mean ICC values obtained were excellent in most cases, and at least good in all cases, for the measurements both without compensation correction and with compensation correction. These values were also higher than those reported by the systematic review (*Williams et al., 2010*) for the goniometer (see Table 1), the instrument that is most typically used in clinical CROM assessment.

Regarding the measurements without compensation correction in the intra-rater analysis, the ICC values of rater B for all movements was higher than those of rater A, although in both cases these results were between excellent and good. In our protocol, the raters did not provide the subjects with any instructions on how to perform the CROM assessment movements; all these instructions were provided by the VR experience, and were therefore identical regardless of which rater was assessing the subject. The assistance of the raters was limited to the placement and usage of the VR equipment. Furthermore, the order of evaluation was randomized. Therefore, the difference in ICC values between the raters must have arisen from the support provided with the VR setup. More specifically, we believe that rater A tended to attach the HMD to the subjects' heads less firmly, which increased the HMD slippage during the movements.

In the case of the inter-rater analysis, it is noteworthy that the ICC values for the second movement of each rater were significantly higher (increase of 0.14 in all the cases, see Table 5) than those for the first movement. A CROM assessment study (*Chalimourdas et al.,*

*2021*) in which inertial sensors placed on the subject's head were used for measurements also reported that the ICC increased when the first test was not considered. The authors of that study attributed this effect to both the raters' and subjects' lack of familiarity with the CROM assessment device. In our case, both raters were quite familiar with VR equipment, so we do not think this factor affected the results. However, the vast majority of the subjects were not familiar with using VR equipment. Wearing a device on the head (the HMD) could have affected how the users performed the movement the first time, since they were not familiar with the feeling. From the second assessment onwards, the users already had some experience with the HMD (since they had performed the warm-up exercises and a complete series of assessment movements before) and we expected them to perform the movements more confidently. Under this hypothesis, the ICC values in Table 4 could be an underestimation of the true ICC values once the users were familiar with the HMD, in the same way that the ICC values in Table 5 are higher for the second measurement. This effect deserves additional study.

The protocol followed in this study was designed under the hypothesis that the tracker located on the back would improve the accuracy of the measurements by allowing the correction of the compensatory movements of the subjects. However, the results of the measurements with compensation correction do not support (at least not completely) this hypothesis. According to Tables 4 and 5, both the intra-rater and inter-rater ICC were lower in flexion and rotation when the compensation correction was applied. However, the intra-rater and inter-rater ICC were higher in the flexion movement when the compensation correction was applied. For the flexion and rotation movements, the compensatory correction was approximately of 8° and 5°, respectively, and during lateral flexion the compensation slightly exceeded 10° (see Table 3). The larger compensation value in the lateral flexion was expected since this movement often feels restrictive and can easily be compensated by an ipsilateral trunk lateral flexion motion.

These compensation magnitudes explain why the ICC decreased for all movements, except for lateral flexion, when the correction was applied. For the flexion and rotation movements, *Measurement_error* was larger than *Trunk_compensation*, while for the lateral flexion the opposite was true (see Eqs. (1) and (3)). Given that the correction of the tracker was counterproductive for two of the movements, and in the case of the other movement the improvement was relatively small (see Table 4 and 5), it is doubtful that the additional step of using the tracker for CROM assessment in healthy subjects is worth it. In the case of subjects with neck pain, their movements could be accompanied by a greater trunk compensation, and therefore the tracker may provide more value.

We are not aware of a previous study that uses a VR-based setup to calculate SEM and MDC. However, we can compare our results with the SEM and MDC results of the study of *Carmona-Pérez et al. (2020)*, who used inertial measurement units (IMU) in CROM assessment. The IMU's SEM values were 7.9° for flexion-extension, 5.4° for right-left rotation, and 5.0° for right-left lateral flexion. The IMU's MDC values were 18.4° for flexion-extension, 12.6° for right-left rotation, and 11.4° right-left lateral flexion (*Carmona-Pérez et al., 2020*). The SEM and MDC obtained in our study, reported in Tables 4 and 5, were lower across all the movements.

The VR system obtained an average SUS score of 86, which is considered excellent usability. In the answers to question 3 ("I thought the system was easy to use") 23 subjects gave this question a score of 5 points ("strongly agree"), six subjects gave it a score of 4 ("agree"), and only one subject gave it a score of 3 ("neutral"). These results suggest that if the subjects are familiar with the VR setup and experience, it would be feasible to perform an autonomous CROM assessment. This could be part of a rehabilitation protocol that combines face-to-face assessment sessions and exercises in a clinic, with remote assessment sessions and exercises from the home of a person with neck pain (*Freimann, Merisalu & Pääsuke, 2015*). This opens the door to using game dynamics techniques to engage the person with neck pain in their rehabilitation process (*Viglialoro et al., 2020*; *Sánchez-Herrera-Baeza et al., 2020*).

### Limitations

This is the first study to investigate a novel VR experience that could assist clinicians in the evaluation of people with neck pain or associated disorders. However, this study has several limitations. The results were limited to the characteristics of the study population, which included asymptomatic, mostly young adult Caucasian participants from a university community. Although this sample may be appropriate to evaluate the general usability and reliability of the VR system in the evaluation of CROM, future research should also investigate whether the results could be influenced by age, previous experience with digital technology, or the presence of craniocervical pain conditions.

A limitation that could affect the repeatability of the study is that during data processing, we made a manual selection of the beginning and end of each movement, although the subsequent calculation of the CROM across different planes from these positions was automatic. This could be improved by developing an algorithm that automatically detects the beginning and end of the movements.

Although the usability and reliability of the VR system ranged between good and excellent, future studies should also evaluate its validity compared to other established CROM measurement instruments.

## CONCLUSION

The VR application developed in this study showed excellent usability and good to excellent intra-rater and inter-rater reliability values in a sample of 30 asymptomatic participants. Therefore, it is feasible to use a VR setup for the CROM assessment without any additional position tracking system. When using a tracker located on the trunk to correct compensatory movements, the ICC values slightly worsened, except in the case of lateral flexion movement, where they slightly improved. Further research is needed to evaluate the validity of using a VR setup for CROM assessment in populations with neck disorders, as well as its potential use in autonomous telematic assessments.

### Funding

This research was funded by the Ministry of Science, Innovation, and Universities of Spain and by the European Regional Development Fund of the European Commission; grant numbers RTI2018-095324-B-I00 and RTI2018-097122-A-I00. The funders had no role in study design, data collection and analysis, decision to publish, or preparation of the manuscript.

### Grant Disclosures

The following grant information was disclosed by the authors:
Ministry of Science, Innovation, and Universities of Spain.
European Regional Development Fund of the European Commission: RTI2018-095324-B-I00, RTI2018-097122-A-I00.

### Competing Interests

The authors declare there are no competing interests.

### Author Contributions

- Jose Angel Santos-Paz conceived and designed the experiments, performed the experiments, analyzed the data, prepared figures and/or tables, authored or reviewed drafts of the article, virtual reality experience development, and approved the final draft.
- Álvaro Sánchez-Picot performed the experiments, analyzed the data, prepared figures and/or tables, authored or reviewed drafts of the article, virtual reality experience development, and approved the final draft.
- Ana Rojo performed the experiments, authored or reviewed drafts of the article, virtual reality experience development, and approved the final draft.
- Aitor Martín-Pintado-Zugasti conceived and designed the experiments, authored or reviewed drafts of the article, and approved the final draft.
- Abraham Otero conceived and designed the experiments, analyzed the data, authored or reviewed drafts of the article, and approved the final draft.
- Rodrigo Garcia-Carmona conceived and designed the experiments, analyzed the data, authored or reviewed drafts of the article, and approved the final draft.

### Human Ethics

The following information was supplied relating to ethical approvals (*i.e.*, approving body and any reference numbers):

The protocol followed in this study was approved by the Universidad San Pablo-CEU Research Ethics Committee.

### Data Availability

The Transcript of the audio message played in the VR application in both its long and short versions, CSV files containing the HMD and tracker measurements for each of the 30

subjects' assessments, and the Matlab script used to obtain the CROM values are available at Github:

https://github.com/jsantospaz/uspceu_crom_vr_assessment.

## Supplemental Information

Supplemental information for this article can be found online at http://dx.doi.org/10.7717/peerj.14031#supplemental-information.

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
