# Peer review of "A novel virtual reality application for autonomous assessment of cervical range of motion: development and reliability study"

_PeerJ, doi:10.7717/peerj.14031_

## Round 0.1 · original submission · Major Revisions

The research has several aspects that must be directed and explained by the authors, among which the research report stands out (e.g. consistency of the concepts throughout the manuscript, length of the introduction, among others), the reproducibility of the methods (eligibility criteria, sample size calculation, among others) and the gap in knowledge that the research attempts to solve. In relation to this last point, it is necessary that the authors are able to show in the introduction in a precise and summarized way "what is not known" about the subject, considering that there is enough literature on the matter:

Sarig-Bahat H, Weiss PL, Laufer Y. Cervical motion assessment using virtual reality. Spine (Phila Pa 1976). 2009;34(10):1018-1024. doi:10.1097/BRS.0b013e31819b3254

Kiper P, Baba A, Alhelou M, et al. Assessment of the cervical spine mobility by immersive and non-immersive virtual reality. J Electromyogr Kinesiol. 2020;51:102397. doi:10.1016/j.jelekin.2020.102397

For more details, please review the comments of the three reviewers.

·

Basic reporting

I suggest a revision by an English proofreading service to improve its writing and comprehension. It provides sufficient context, with minimal details. Please check my comments for further understanding.

Experimental design

There are sensitive points that could affect the reproducibility of the results. Please check my comments for further understanding.

Validity of the findings

There are some sensitive points that should be clarified in order to consider their reproducibility. The authors should clarify some of the procedures performed to achieve a high level of robustness. Regarding conclusions, these should be more precise. Please check my comments for further understanding.

Additional comments

No comment

Reviewer 2 ·

Basic reporting

The article is deficient in basic aspects of the analysis and description of human movement and does not include sufficient background to support the relevance of the work in the broader field of knowledge. It is recommended to review definitions related to the spatial description of human movement, as well as the theoretical framework of compensatory motor strategies.

The author responds to hypotheses that were not adequately presented and substantiated in the introduction or methods used.

Table 3 can be improved considering the potential interest of the work in clinical contexts, differentiating the range of motion in cervical flexion and extension, as well as flexion and left/right rotation.

Experimental design

It is requested to explain in the methodology how it was ensured that people perform the movement of the neck in its maximum active range of motion.

The standard of the procedure used to determine trunk compensation is not robust, therefore it is recommended to reconsider the conclusions linked to these data.

Validity of the findings

Avoid citing tables and figures in the conclusion and limit it to the data obtained in the evaluated population, healthy people in this case.

Pointing out the indications given by the VR system to the participants in order to perform the neck movements could contribute to the reproducibility of the protocol.

Additional comments

no comment

Annotated reviews are not available for download in order to protect the identity of reviewers who chose to remain anonymous.

Reviewer 3 ·

Basic reporting

No comment

Experimental design

All information about configuration and usage description VR should be taken from the introduction and added in the methods section, instrumentation subsection.

The description of the warm-up appears repeated in methods.

Information on sample characterization is missing. The authors mention that they were young university students only in the limitations of the study

Validity of the findings

The authors' conclusion is very extensive, with information on results and literature. I suggest authors to put this information at the end of the discussion and redo the conclusion according to the study questions

Additional comments

Correct the sentence on page 14, line 558 " These values were expected since the lateral flexion movement is more awkward and it is often accompanied by trunk movements”

---

## Round 0.2 · Major Revisions

The authors have made a great effort to satisfy the demands and corrections of the reviewers, reflected in the improvement of the research report. However, based on the reviewers' suggestions, the manuscript needs to be edited by an English proofreading service to improve its writing and comprehension. This aspect is essential to comply with PeerJ standards.

·

Basic reporting

I suggest a revision by an English proofreading service to improve its writing and comprehension.

Experimental design

No comment

Validity of the findings

No comment

Additional comments

I would like to congratulate the authors of the manuscript for the revisions made and for clarifying every point of query I raised

Reviewer 2 ·

Basic reporting

NO COMMENT

Experimental design

NO COMMENT

Validity of the findings

NO COMMENT

Additional comments

NO COMMENT

---

## Round 0.3 · accepted · Accept

The authors have complied with the suggestions and corrections of the reviewers, reflected in the improvement of the research report.